# Coarse and Fine-grained Forecasting Via Gaussian Process Blurring Effect

## Abstract

Time series forecasting is a challenging task due to the existence of complex and dynamic temporal dependencies, leading to inaccurate predictions even by the most advanced models. While increasing training data is a common approach to enhance accuracy, it is often a limitted source. In contrast, we are building on successful denoising approaches for image generation by proposing an end-to-end forecast-blur-denoise framework. By training the parameters of the blur model for best end-to-end performance, we advocate for a clear division of tasks between the forecasting and denoising models. This encourages the forecasting model to learn the coarse-grained behavior, while the denoising model is filling in the blurred fine-grained details. Our experiments show that our proposed approach is able to improve the forecasting accuracy of several state-of-the-art forecasting models as well as several other denoising approaches. The code for reproducing our main result is open-sourced and available online.[1]

## 1 Introduction

Time series forecasting is a vital foundational technology in many important domains such as in economics Capistrán et al. (2010), health care Lim (2018), demand forecasting Salinas et al. (2020) and autonomous driving Chang et al. (2019). Despite the recent advances in neural networks, time series forecasting still remains a challenging problem. The complexity and dynamic nature of temporal dependencies on different time scales pose a challenge for forecasting models to accurately resemble the temporal patterns of the target variable.

Denoising models Ho et al. (2020) have recently gained popularity in generating high quality images. Typically denoising models Vincent et al. (2010) are trained to reverse an image corrupting process, thereby learning correlation patterns of the application domain. In this work, we rethink ideas from denoising models in applications to time series forecasting task, which is defined as follows:

**Time series forecasting task.** We define a time series as a sequence of observations $\{\boldsymbol{\chi}_t, \boldsymbol{\gamma}_t\}_t$ over time steps $t$. The forecasting task involves predicting values of the target variable $\boldsymbol{\gamma}$ for the next $\tau$ time steps into the future (from $t_0$ to $t_0 + \tau$), given historical $\kappa$ time series observations prior to a cutoff time step $t_0$. We refer to the given historical time series as $X = \{\boldsymbol{\chi}_t, \boldsymbol{\gamma}_t\}_{t=t_0-\kappa}^{t_0}$, and the to be predicted as $Y = \{\boldsymbol{\gamma}_t\}_{t=t_0}^{t_0+\tau}$. Each variable $\boldsymbol{\chi}_t$ consists of $d_\chi$ features including 1) time-based features such as time of the day, season of the year, and 2) other stochastic features observed at time step $t$. Each target variable $\boldsymbol{\gamma}_t$ consists of $d_\gamma \geq 1$ variable(s) of interests, yieldings $X_t \in \mathbb{R}^{d_\chi + d_\gamma}$ and, $Y_t \in \mathbb{R}^{d_\gamma}$.

**Idea.** Inspired by the success of denoising models in generating high quality images by reverting a noise process, our approach involves constructing improved coarse and fine-grained forecasting models.

We introduce an innovative forecasting and denoising framework by presenting a forecast-blur-denoise approach. In our approach, we introduce a division of labor by utilizing a trainable blur/noise model that

---

[1]Code available at: https://anonymous.4open.science/r/Fine_grained_Gaussian_Process_Forcasting-4E36

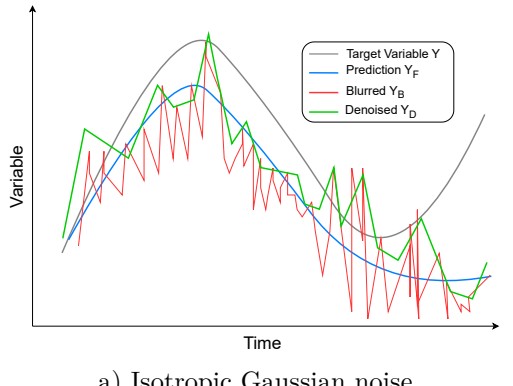 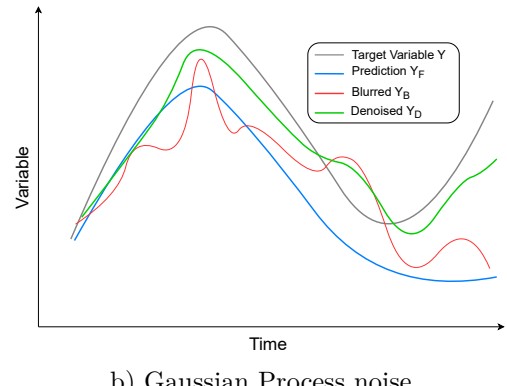

a) Isotropic Gaussian noise b) Gaussian Process noise

Figure 1: A synthetic example of blurring and denoising the prediction. The left figure a) illustrates that blurring and denoising the prediction with isotropic Gaussian noise results in less desirable forecasts, where the denoising model attempts to remove the jitters. However, as depicted in the right figure b) blurring and denoising the prediction with GP model results in a smooth behavior with improved fine-grained details.

distributes responsibilities between the forecasting and denoising models. By training the parameters of the blur model end-to-end with the forecasting and denoising model, we encourage the forecasting model to accurately predict the broad and coarse-grained behavior (as fine-grained features will be blurred out), while the denoising model is filling the fine-grained details blurred by the blur model.

When applied to image generation, denoising models usually revert a corruption process of pixel-wise isotropic Gaussian noise. With applications to time series forecasting, however, the isotropic corruption process will introduce what we call *jitters* (see red function in Figure 1 left), as the noise added to one time step is independent of noise added to subsequent time steps. Since most modern time series forecasting models produce forecasts with smooth behavior, the process of adding isotropic noise yields non-smooth time series that are rather unnaturally behaving. Hence, we hypothesize that the benefit of isotropic noise for improving time series forecasting is somewhat limited, as it prompts the denoising model to merely eliminate the introduced jitters.

In this work we explore alternative techniques for training denoising models for time series forecasting. As our goal is to train the denoising model to focus on accurately predicting the fine-grained details (rather than removing jitters), we are interested in a blur model that locally blurs the initial forecasts by generating smooth and correlated signals. Given that the isotropic Gaussian will generate jitters due to its temporally uncorrelated nature, in line with Robinson et al. (2018) we will employ a Gaussian Process (GP) as the blur model that naturally models correlation across time to provide smooth functions (see red function in Figure 1 right). By end-to-end training the parameters of the GP model, we attempt a division of labor between forecasting and denoising models. We hypothesize that the "right" blur model is essential for competence partitioning to achieve improved coarse and fine-grained forecasting. This hypothesis is supported in the experimental section.

**Related work.** Our goal is to improve the forecasting ability of the existing time series forecasting models. Among various time series forecasting models, the ones that utilize transformers have demonstrated superior performance (Li et al., 2019; Fan et al., 2019). But even the state-of-the-art time series forecasting models make wrong predictions. We will apply our approach to improve two of the best forecasting models, the Autoformer and the Informer model. The Autoformer model Wu et al. (2021) improves on the basic attention mechanism by decomposing a time series into sub-series and incorporating an auto-correlation mechanism to capture the correlation between these sub-series. This leads to gains in efficiency and accuracy. The Informer model Zhou et al. (2021) employs ProbSparse attention, which prunes the attention matrix by focusing on samples that are outliers from a uniform distribution, resulting in enhanced efficiency and accuracy. Both these models are included in our experimental evaluation.

Probabilistic time series forecasting models such as DeepAR Salinas et al. (2020) generate predictions by drawing samples from a learned Isotropic Gaussian distribution. These models aim to learn the parameters of the Gaussian distribution via a deep neural network model such as LSTMs Hochreiter & Schmidhuber (1997).

Denoising models often revert a corruption process applied to the input during training to increase the robustness and generalization of the model. Therefore, the performance of time series forecasting models could as well be improved when integrated with a denoising approach. One of such approaches is the TimeGrad Rasul et al. (2021) forecasting model that uses denoising diffusion models to reverse the isotropic Gaussian noise added to the input time series. TimeGrad estimates the parameters of the Gaussian distribution using recurrent neural networks architectures LSTM/GRU. However, prior works including Koohfar & Dietz (2022); Khandelwal et al. (2018) show that LSTMs do not perform competitively compared to transformer-based forecasting methods.

DLinear model proposed by Zeng et al. (2023) proposes a *navive* time series forecasting model that only uses simple multi-layer perceptrons projections from previous observations to make predictions. We compare our proposed model to DLinear model in our experimental section.

Recently, Li et al. (2022) introduce D3VAE, a novel approach that combines bidirectional variational auto-encoder techniques with diffusion, denoising, and disentanglement to enhance time series representation. However, in line with DLinear, this approach does not include sequential modeling techniques, resulting in limited competitiveness when compared to transformer-based forecasting methods as shown in our experimental section.

**Contributions.**   In this work, we take a different approach by enhancing the performance of forecasting models by proposing an enhanced coarse and fine-grained forecasting framework. (1) Rather than blurring the input solely during training, we advocate for an end-to-end forecast-blur-denoise forecasting framework that encourages a separation of concerns for the forecasting and denoising models. (2) Our complementary approach can be readily added to a wide-range of forecasting models. (3) Our novel approach is an alternative towards approaches that use denoising solely during training or when leveraging boosting. We experimentally show that our approach predominately outperforms many of those baselines. (4) We further demonstrate that for smooth time series data, isotropic Gaussians are not a suitable noise model.

## 2 Methodology

Time series forecasting models using a ground truth series $Y$, are trained to learn the expected behavior of the target variable over time. However, complex dependencies can lead to erroneous predictions. Adding more training data can help to improve the forecasting model's performance, but it is often a limited resource. In this work we explore how ideas from denoising models can improve the performance of the family of forecasting models.

### 2.1 Background: Isotropic Denoising Approaches

Denoising models are trained to reverse a corruption process. For this purpose, true data $X$ is corrupted with noise to obtain a corrupted version $\tilde{X}$. The denoising part of the model is trained to, given $\tilde{X}$, predict the original data $X$. Often this is modelled by predicting the noise, which is then subtracted from $\tilde{X}$ to obtain the original data $X$. Denoising models can be trained for the iterative reversal of noise (reasoning on a series of latents with different noise level) or in single step with a conditional model $X|\tilde{X}$ (Deja et al., 2022). The denoising objective can be optimized by itself or in combination with other objectives such as prompted image generation or time series forecasts (Nichol & Dhariwal, 2021).

Most current work uses a simple corruption process that employs an isotropic Gaussian noise model, where $\tilde{X} \sim \mathcal{N}(\tilde{X}; X, \sigma^2\mathbf{I})$ that acts independently on different data points, i.e., pixels for image generation Nichol & Dhariwal (2021) or time steps for time series forecasts (Rasul et al., 2021).

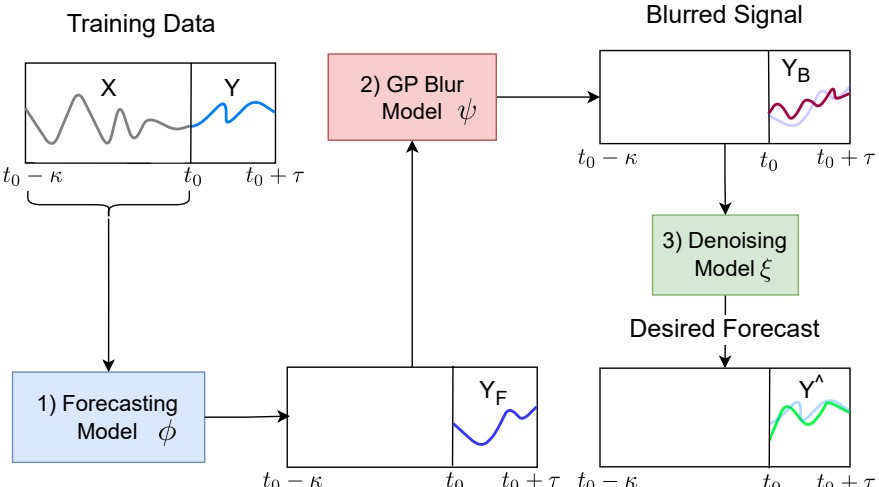

Figure 2: Our proposed model's framework for end-to-end training the forecasting, GP blur, and denoising model. The multi-step neural network forecasting model predicts $Y_F$ from historical observations $X$. The predictions $Y_F$ are then blurred by a trained GP model to obtain $Y_B$. The blurred predictions $Y_B$ are then denoised by our denoising model to obtain $Y_D = \hat{Y}$.

## 2.2 Issues with Uncorrelated Noise Models for Time Series

Isotropic Gaussian noise is one of the most commonly employed corruption processes, which provides noise that is identically and independently distributed (*i.i.d.*) and when used to generate observations for time series are lacking smooth behavior over time. While isotropic Gaussian noise corruption can (and are) applied to time series, the result is a corruption in the form of jitters (as depicted in red in Figure 1a). However, time series forecasting models are based on the assumption that data points are correlated over time—and hence not i.i.d. The effect on the training problem is that the denoiser may only learn to remove jitters. However, most errors in forecasting models are not due to jitters, as predicted forecasts are usually smooth functions that merely exhibit incorrect fine-grained behavior. In this work we go even one step further and train a smooth and correlated noise/blur model. We hypothesize that this model is more advantageous as it directs the denoising model to focus on accurately predicting the fine-grained details that are locally blurred, rather than simply eliminating jitters.

## 2.3 Locally-blurred Signals with Gaussian Processes

To obtain an improved fine-grained forecasting model, we instead propose to include a noise/blur model that generates smooth temporally-correlated functions, to train the denoising model, as depicted in Figure 1b). We employ a Gaussian Process (GP) which models the correlation between consecutive samples in a sequence of observations via a kernel function $k_\psi$.

We generate smoothly correlated signals by drawing samples from the probability density function (PDF) of Gaussian Process (GP) denoted as $b_\psi$:

$$b_\psi(\tilde{X}|X) = \mathcal{N}(\tilde{X}; X, k_\psi(X, X) + \sigma^2\mathbf{I}) \tag{1}$$

where $\psi$ denotes the parameters of the GP kernel that are trained as part of our proposed joint model for the best end-to-end performance.

In Section 3 we will experimentally support our claim that locally blurring the initial forecasts with GPs leads to more effective training for time series forecasting than isotropic Gaussians.

### 2.4 Forecast-blur-denoise Forecasting Framework

There are many ways to exploit ideas from denoising models for time series forecasting: as separate negative training data, as a secondary objective on true time series, via auto-encoders, or as an integral part of an end-to-end model. Based on preliminary experiments, in this work we integrate ideas into a joint forecast-blur-denoise forecasting model. The model integrates 1) a forecasting model, 2) a GP blur model, and 3) a denoising model as depicted in Figure 2. All parts are jointly trained for best MSE performance. We describe these components in detail below.

**1) Forecasting model:** Any time series forecasting model can be used here that, given the observations $X = \{\boldsymbol{x}_t; \boldsymbol{\gamma}_t\}_{t=t_0-\tau}^{t_0}$ predicts the future target variables $Y_F$, as represented by the blue box in Figure 2. We experimentally demonstrate that our end-to-end framework will help train enhanced coarse and fine-grained forecasting models. We refer to the set of parameters of the forecasting model as $\phi$.

**2) GP blur model:** The initial predictions $Y_F$ are locally blurred by the mean function of the GP model, depicted as a light red box in Figure 2. As described above, we suggest to use GP as the blur model to obtain $Y_B \sim b_\psi(Y_B|Y_F)$. GP parameters $\psi$ are trained jointly with other parameters of our end-to-end forecasting and denoising model. Alternative noise models could be used here, which we explore in Section 3.

**3) Denoising Model:** Given the locally blurred predictions $Y_B$, the denoising model focuses on predicting the fine-grained details by removing the blurs, which consequently improves the initial forecasting. While many architectures could be chosen for the denoising model, we choose to use the same time series forecasting model with a new set of parameters $\xi$ as the denoiser to obtain final predictions $Y_D = \hat{Y}$. The denoising model is represented by the green box in Figure 2.

The result is a compound model that encourages the initial forecasting model to focus on modelling coarse-grained behavior, and a denoising model that corrects the fine-grained details. This is encouraged by the GP model that will "blur" fine-grained details in the forecast, and a denoising model that focuses on correcting these fine-grained details. Additionally, the denoising component acts as a fall-back for when the initial forecasting model fails, reducing the likelihood of catastrophic errors.

Note that for fixed training data $X$ and $Y$, a new blurring effect is sampled in every epoch, deterring the model from overfitting to any particular blurring pattern.

To optimize the GP model's parameters, we employ a strategy reminiscent of the scalable variational GP method introduced by Hensman et al. (2015). Their scalable variational GP technique offers a computationally efficient approximation of the GP model, achieving nearly linear computational complexity for $k_\psi(X, X)$ as the forecasting horizon increases.

### 2.5 End-to-end Forecasting and GP Loss

With an abundance of training data, the compound model could be trained end-to-end, predicting $\hat{Y}$ from given $X$ and minimizing the distance to the ground truth $Y$ via an MSE (or $L_2$) forecasting loss.

We optimize the parameters of our GP model using the ground truth $Y$. This allows for the efficient optimization of the parameters of variational GPs. The compound loss function employed for end-to-end training is defined as follows, where the variational evidence lower bound (ELBO) optimizes the GP model to obtain an ideal blurring effect:

$$\mathcal{L} = \underbrace{L_{\texttt{MSE}}}_{\text{forecasting loss}} (\hat{Y} = Y|X, Y_F, Y_B, \phi, \psi, \xi) + \underbrace{\lambda L_{\texttt{ELBO}}(Y_B = Y|X, Y_F, \phi, \psi)}_{\text{GP loss}} \qquad (2)$$

In our experiments, following Nichol & Dhariwal (2021) we set $\lambda$ to a small number ($\lambda = 0.001$) to prevent the loss $L_{\texttt{ELBO}}$ from overwhelming the $L_{\texttt{MSE}}$ loss. Figure 2 illustrates our end-to-end framework.

Table 1: Overall results of the quantitative evaluation of our forecast-blur-denoise and other baseline models in terms of average and standard error of **MSE**. We compare the forecasting models on all three datasets with different number of forecasting steps. A lower **MSE** indicates a better model. Our forecast-blur-denoise with GPs enhances the performance of the original Autoformer and isotropic Gaussian noise model (AutoDI). Note that for a fair comparison, all baseline models share the same experimental setup as our proposed model. Reported results may differ from the original baseline papers, and the baseline models are available in our online repository.

| Dataset | Horizon | **AutoDG(Ours)** | Autoformer | AutoDI | NBeats | DLinear | DeepAR | CMGP | ARIMA |
|---|---|---|---|---|---|---|---|---|---|
| Traffic | 24 | **0.392** | 0.412 | 0.405 | 0.475 | 0.553 | 0.888 | 0.824 | 1.436 |
| | | ±0.006 | ±0.006 | ±0.003 | ±0.008 | ±0.000 | ±0.000 | ±0.000 | ±0.000 |
| | 48 | **0.387** | 0.422 | 0.416 | 0.462 | 0.547 | 0.944 | 0.828 | 1.444 |
| | | ±0.001 | ±0.004 | ±0.001 | ±0.012 | ±0.000 | ±0.000 | ±0.000 | ±0.000 |
| | 72 | **0.380** | **0.383** | 0.394 | 0.465 | 0.540 | 0.877 | 0.893 | 1.459 |
| | | ±0.001 | ±0.003 | ±0.002 | ±0.003 | ±0.000 | ±0.000 | ±0.000 | ±0.000 |
| | 96 | **0.385** | 0.400 | 0.411 | 0.464 | 0.539 | 0.860 | 0.859 | 1.444 |
| | | ±0.003 | ±0.004 | ±0.002 | ±0.002 | ±0.000 | ±0.000 | ±0.000 | ±0.000 |
| Electricity | 24 | **0.165** | 0.187 | 0.170 | 0.200 | 0.222 | 1.039 | 1.000 | 1.707 |
| | | ±0.001 | ±0.003 | ±0.001 | ±0.001 | ±0.000 | ±0.000 | ±0.000 | ±0.000 |
| | 48 | **0.188** | 0.203 | 0.207 | 0.218 | 0.238 | 1.014 | 0.987 | 1.729 |
| | | ±0.003 | ±0.008 | ±0.003 | ±0.000 | ±0.000 | ±0.000 | ±0.000 | ±0.000 |
| | 72 | **0.209** | 0.230 | 0.253 | 0.234 | 0.264 | 1.023 | 0.993 | 1.759 |
| | | ±0.004 | ±0.001 | ±0.004 | ±0.007 | ±0.000 | ±0.000 | ±0.000 | ±0.000 |
| | 96 | **0.211** | 0.230 | 0.316 | 0.237 | 0.264 | 1.013 | 0.971 | 1.747 |
| | | ±0.001 | ±0.014 | ±0.002 | ±0.001 | ±0.000 | ±0.000 | ±0.000 | ±0.000 |
| Solar | 24 | **0.446** | 0.472 | 0.473 | 0.612 | 0.828 | 0.999 | 1.001 | 1.869 |
| | | ±0.002 | ±0.003 | ±0.001 | ±0.006 | ±0.000 | ±0.000 | ±0.000 | ±0.000 |
| | 48 | **0.546** | 0.603 | 0.574 | 0.717 | 0.928 | 0.968 | 1.007 | 1.872 |
| | | ±0.003 | ±0.004 | ±0.001 | ±0.001 | ±0.000 | ±0.000 | ±0.000 | ±0.000 |
| | 72 | **0.666** | **0.667** | 0.698 | 0.766 | 0.978 | 0.974 | 1.002 | 1.855 |
| | | ±0.003 | ±0.004 | ±0.002 | ±0.006 | ±0.000 | ±0.000 | ±0.000 | ±0.000 |
| | 96 | **0.713** | 0.739 | 0.730 | 0.827 | 1.004 | 0.974 | 0.997 | 1.874 |
| | | ±0.004 | ±0.009 | ±0.005 | ±0.001 | ±0.000 | ±0.000 | ±0.000 | ±0.000 |

## 3 Experiments

We experimentally demonstrate the success of our forecast-blur-denoise approach across three datasets and two state-of-the-art forecasting models. We first focus on demonstrating the efficacy of our treatment and the importance of using correlated noise of GPs, rather than isotropic Gaussians. Next, we compare our denoising approach to a wide range of canonical denoising and ensemble baselines.

### 3.1 Experimental Setup

We lay out the conditions for our experimental evaluation.

**Datasets.** We select three widely used datasets that have been used for training and validation by a significant amount of research papers (Salinas et al., 2020; Li et al., 2019; Wu et al., 2021; Zhou et al., 2021).

**Traffic:**[2] A univariate dataset, containing the occupancy rate ($y_t \in [0, 1]$) of 440 SF Bay Area freeways, aggregated on hourly interval.

---

[2]URL to Traffic dataset

Table 2: Overall results of the quantitative evaluation of our forecast-blur-denoise and other baseline models in terms of average and standard error of **MSE**. We compare the forecasting models on all three datasets with different number of forecasting steps. A lower **MSE** indicates a better model. Our forecast-blur-denoise with GPs enhances the performance of the original Informer and isotropic Gaussian noise model (InfoDI). Note that for a fair comparison, all baseline models share the same experimental setup as our proposed model. Reported results may differ from the original baseline papers, and the baseline models are available in our online repository.

| Dataset | Horizon | InfoDG(Ours) | Informer | InfoDI | NBeats | DLinear | DeepAR | CMGP | ARIMA |
|---|---|---|---|---|---|---|---|---|---|
| Traffic | 24 | **0.398** | 0.421 | 0.415 | 0.475 | 0.553 | 0.888 | 0.824 | 1.436 |
| | | ±0.006 | ±0.006 | ±0.003 | ±0.008 | ±0.000 | ±0.000 | ±0.000 | ±0.000 |
| | 48 | **0.399** | 0.434 | 0.395 | 0.462 | 0.547 | 0.944 | 0.828 | 1.444 |
| | | ±0.001 | ±0.004 | ±0.001 | ±0.012 | ±0.000 | ±0.000 | ±0.000 | ±0.000 |
| | 72 | **0.380** | 0.436 | 0.395 | 0.465 | 0.540 | 0.877 | 0.893 | 1.459 |
| | | ±0.001 | ±0.001 | ±0.002 | ±0.002 | ±0.000 | ±0.000 | ±0.000 | ±0.000 |
| | 96 | **0.397** | **0.402** | **0.402** | 0.464 | 0.539 | 0.860 | 0.859 | 1.444 |
| | | ±0.003 | ±0.003 | ±0.004 | ±0.004 | ±0.000 | ±0.000 | ±0.000 | ±0.000 |
| Electricity | 24 | **0.193** | 0.222 | 0.212 | 0.200 | 0.222 | 1.039 | 1.000 | 1.707 |
| | | ±0.001 | ±0.001 | ±0.003 | ±0.001 | ±0.000 | ±0.000 | ±0.000 | ±0.000 |
| | 48 | **0.222** | 0.262 | 0.229 | 0.218 | 0.238 | 1.014 | 0.987 | 1.729 |
| | | ±0.003 | ±0.007 | ±0.003 | ±0.003 | ±0.000 | ±0.000 | ±0.000 | ±0.000 |
| | 72 | 0.238 | 0.280 | 0.253 | **0.234** | 0.264 | 1.023 | 0.993 | 1.759 |
| | | ±0.001 | ±0.004 | ±0.004 | ±0.007 | ±0.000 | ±0.000 | ±0.000 | ±0.000 |
| | 96 | 0.242 | 0.289 | 0.275 | **0.237** | 0.264 | 1.013 | 1.130 | 1.747 |
| | | ±0.001 | ±0.002 | ±0.014 | ±0.001 | ±0.000 | ±0.000 | ±0.000 | ±0.000 |
| Solar | 24 | **0.455** | 0.524 | **0.465** | 0.612 | 0.828 | 0.999 | 0.971 | 1.869 |
| | | ±0.009 | ±0.003 | ±0.006 | ±0.006 | ±0.000 | ±0.000 | ±0.000 | ±0.000 |
| | 48 | **0.556** | 0.629 | 0.570 | 0.717 | 0.928 | 0.968 | 1.007 | 1.872 |
| | | ±0.005 | ±0.003 | ±0.005 | ±0.001 | ±0.000 | ±0.000 | ±0.000 | ±0.000 |
| | 72 | **0.643** | 0.729 | 0.707 | 0.766 | 0.978 | 0.974 | 1.002 | 1.855 |
| | | ±0.003 | ±0.023 | ±0.002 | ±0.006 | ±0.000 | ±0.000 | ±0.000 | ±0.000 |
| | 96 | **0.708** | 0.770 | 0.766 | 0.827 | 1.004 | 0.974 | 0.997 | 1.874 |
| | | ±0.004 | ±0.004 | ±0.009 | ±0.005 | ±0.000 | ±0.000 | ±0.000 | ±0.000 |

**Solar Energy:**[3]: A univariate dataset about solar power that could be obtained across different locations in America, collected on an hourly interval.

**Electricity:**[4] A univariate dataset listing the electricity consumption of 370 customers, aggregated on an hourly level.

From each dataset, we roughly use 40,000 samples, where each sample contains given observations $X$ of $\kappa = 192$ time steps, from which (using multiple horizon forecasting) we predict the next $\tau \in \{24, 48, 72, 96\}$ future time steps. After Z-score normalization, we partition 40,000 samples of each dataset into three parts, 80% for training, 10% for validation, and 10% for performance evaluation.

**Evaluation metrics.** We evaluate our model and other alternatives using mean squared error MSE $= \frac{1}{n}(\sum_{t=1}^{n}(\boldsymbol{y}_t - \hat{\boldsymbol{y}}_t)^2)$, where $n$ denotes the length of the predicted time series. We also study the mean absolute error MAE $= \frac{1}{n}(\sum_{t=1}^{n}|\boldsymbol{y}_t - \hat{\boldsymbol{y}}_t|)$, where we obtain the same findings (omitted from this manuscript, but available in Appendix A).

---

[3]URL to Solar dataset

[4]URL to Electricity Dataset

Table 3: Comparison of different denoising baselines to our forecast-blur-denoise approach with GPs when treating Autoformer forecasting model. We find that our approach predominantly outperforms the other denoising approaches. Results are reported as average and standard error of **MSE**. A lower **MSE** indicates a better forecasting model.

| Dataset | Horizon | **AutoDG(Ours)** | Autoformer | AutoDI | AutoDWC | AutoRB | AutoDT |
|---|---|---|---|---|---|---|---|
| Traffic | 24 | **0.392** ±0.006 | 0.412 ±0.006 | 0.405 ±0.003 | 0.400 ±0.005 | 0.447 ±0.006 | 0.430 ±0.015 |
| | 48 | **0.387** ±0.001 | 0.422 ±0.007 | 0.416 ±0.007 | 0.417 ±0.009 | 0.450 ±0.005 | 0.410 ±0.005 |
| | 72 | **0.380** ±0.001 | **0.383** ±0.002 | 0.394 ±0.002 | 0.398 ±0.003 | 0.430 ±0.004 | 0.404 ±0.006 |
| | 96 | **0.385** ±0.003 | 0.400 ±0.004 | 0.411 ±0.002 | 0.405 ±0.001 | 0.413 ±0.002 | 0.422 ±0.002 |
| Electricity | 24 | **0.165** ±0.001 | 0.187 ±0.003 | 0.170 ±0.001 | 0.174 ±0.00 | 0.260 ±0.001 | 0.170 ±0.007 |
| | 48 | **0.188** ±0.003 | 0.203 ±0.008 | 0.207 ±0.003 | 0.219 ±0.002 | 0.222 ±0.002 | 0.200 ±0.002 |
| | 72 | **0.209** ±0.004 | 0.230 ±0.001 | 0.253 ±0.004 | 0.218 ±0.010 | 0.234 ±0.022 | 0.212 ±0.002 |
| | 96 | **0.211** ±0.001 | 0.230 ±0.014 | 0.316 ±0.002 | 0.226 ±0.008 | 0.296 ±0.011 | 0.218 ±0.004 |
| Solar | 24 | **0.446** ±0.002 | 0.472 ±0.003 | 0.473 ±0.006 | **0.449** ±0.003 | 0.527 ±0.006 | 0.457 ±0.004 |
| | 48 | **0.546** ±0.005 | 0.603 ±0.003 | 0.574 ±0.001 | 0.605 ±0.005 | 0.595 ±0.005 | 0.598 ±0.003 |
| | 72 | **0.666** ±0.003 | **0.667** ±0.004 | 0.698 ±0.002 | 0.690 ±0.010 | 0.718 ±0.002 | 0.670 ±0.006 |
| | 96 | **0.713** ±0.004 | 0.739 ±0.009 | 0.730 ±0.005 | 0.732 ±0.006 | 0.753 ±0.007 | 0.733 ±0.006 |

### 3.2 Treated Time Series Forecasting Models and Baselines

Since our forecast-blur-denoise approach can be used to treat any forecasting model, we study the benefit for the following state-of-the-art time series forecasting models. The number of layers is tuned with Optuna.

**Autoformer (Wu et al., 2021):** A multi-layer Autoformer model with auto-correlation

**Informer (Zhou et al., 2021):** A multi-layer informer with ProbAttention.

We conduct a comparative analysis focusing on the following treatments applied to the Autoformer and Informer forecasting model:

**AutoDG/InfoDG (our proposed model):** Our proposed forecast-blur-denoise framework with GP blur model as described in Section 2.

**AutoDI/InfoDI (denoise scaled Isotropic noise):** the same forecast-blur-denoise model, albeit using scaled isotropic Gaussians as the noise model (instead of the GP).

**Auto(In)former (only forecasting):** the initial untreated forecasting model.

We also compare against the five following baselines:

**1) ARIMA (Hyndman & Khandakar, 2008):** An autoregressive integrated moving average.

**2) CMGP (Chakrabarty et al., 2021):** Model calibration using Bayesian Optimization and GPs.

**3) DeepAR (Salinas et al., 2020):** An Autoregressive probabilistic time series forecasting model that uses LSTMs to estimate the parameters of a Gaussian distribution.

**4) DLinear (Zeng et al., 2023):** A forecasting model that uses multi-layer perceptron to make predictions.

**5) Nbeats (Oreshkin et al., 2020):** A neural approach for interpreting trend, seasonality, and residuals.

Table 4: Comparison of different denoising baselines to our forecast-blur-denoise approach with GPs when treating Informer forecasting model. We find that our approach predominantly outperforms the other denoising approaches. Results are reported as average and standard error of **MSE**. A lower **MSE** indicates a better forecasting model.

| Dataset | Horizon | **InfoDG(Ours)** | Informer | InfoDI | InfoDWC | InfoRB | InfoDT |
|---------|---------|------------------|----------|--------|---------|--------|--------|
| Traffic | 24 | **0.398**±0.005 | 0.421±0.005 | 0.415±0.002 | 0.406±0.002 | 0.435±0.005 | 0.473±0.003 |
|  | 48 | **0.399**±0.004 | 0.434±0.014 | 0.395±0.007 | 0.392±0.003 | **0.395** ±0.011 | 0.421±0.014 |
|  | 72 | **0.380**±0.001 | 0.436±0.015 | 0.395±0.001 | 0.392±0.001 | 0.407±0.009 | 0.421±0.011 |
|  | 96 | **0.397**±0.003 | 0.402±0.002 | **0.402**±0.006 | **0.394**±0.003 | 0.412±0.007 | 0.414±0.015 |
| Electricity | 24 | **0.193**±0.003 | 0.222±0.006 | 0.212±0.001 | 0.204±0.005 | 0.225±0.009 | 0.230±0.007 |
|  | 48 | **0.222**±0.003 | 0.262±0.013 | 0.229±0.003 | 0.241±0.007 | 0.261±0.014 | 0.256±0.004 |
|  | 72 | **0.238**±0.001 | 0.280±0.006 | 0.253±0.006 | 0.263±0.013 | 0.262±0.008 | 0.268±0.008 |
|  | 96 | **0.242**±0.004 | 0.289±0.011 | 0.275±0.005 | 0.279±0.006 | 0.283±0.001 | 0.275±0.007 |
| Solar | 24 | **0.455**±0.007 | 0.524±0.002 | 0.465±0.006 | 0.457±0.006 | 0.498±0.010 | 0.512±0.012 |
|  | 48 | **0.556**±0.011 | 0.629±0.021 | 0.570±0.007 | 0.590±0.016 | 0.623±0.016 | 0.629±0.023 |
|  | 72 | **0.643**±0.022 | 0.729±0.024 | 0.707±0.026 | 0.708±0.014 | 0.748±0.010 | 0.726±0.006 |
|  | 96 | **0.708**±0.010 | 0.770±0.017 | 0.766±0.006 | 0.739±0.010 | 0.781±0.017 | 0.777±0.000 |

In an ablation study, we additionally compare against the following canonical denoising and boosting approaches for the Autoformer and Informer models.

**AutoDWB/InfoDWB (denoise without blur):** a forecast-denoise model, where the denoising acts directly on the predictions (no blurring).

**AutoRB/InfoRB (residual-boosted):** two forecasting models, where the second is trained on minimizing the error residuals between the predictions and the ground-truth.

**AutoDT/InfoDT (denoise only during Training):** the forecasting model is trained to denoise with GP blurring, but the blurring is not used at test time.

### 3.3 Model training and Hyper-parameters

All models are trained and evaluated multiple times using different random seeds. We use Optuna Akiba et al. (2019) for hyper-parameter optimization. We tune the warm up steps of the optimization, model size (dimensionality of latent space) for all models, chosen from {16, 32}, and the number of layers of the forecasting model chosen from {1, 2}. Scaling factor $\sigma$ for the scaled isotropic Gaussian noise model is trained as a model parameter with values between the range of $[0, 0.1]$.

We use 8-head attention for all attention-based models. We model the GP using ApproximateGP of the GPyTorch package. [5] The batch size is set to 256. We use the Adam Kingma & Ba (2015) optimizer, we change the learning rate following Vaswani et al. (2017) with warm-up steps chosen from {1000, 8000}. All models are trained on a single NVIDIA A40 GPU with 45GB of memory. We train our forecasting and denoising model with total number of 50 epochs. Training one epoch of our end-to-end model roughly takes about 25 seconds.

### 3.4 Results and Discussion

Table 1 and 2 summarize the evaluation results of the treatments of the Autoformer and Informer forecasting models along with other baselines on the three datasets. Results are reported as average and standard errors

---

[5] https://gpytorch.ai/

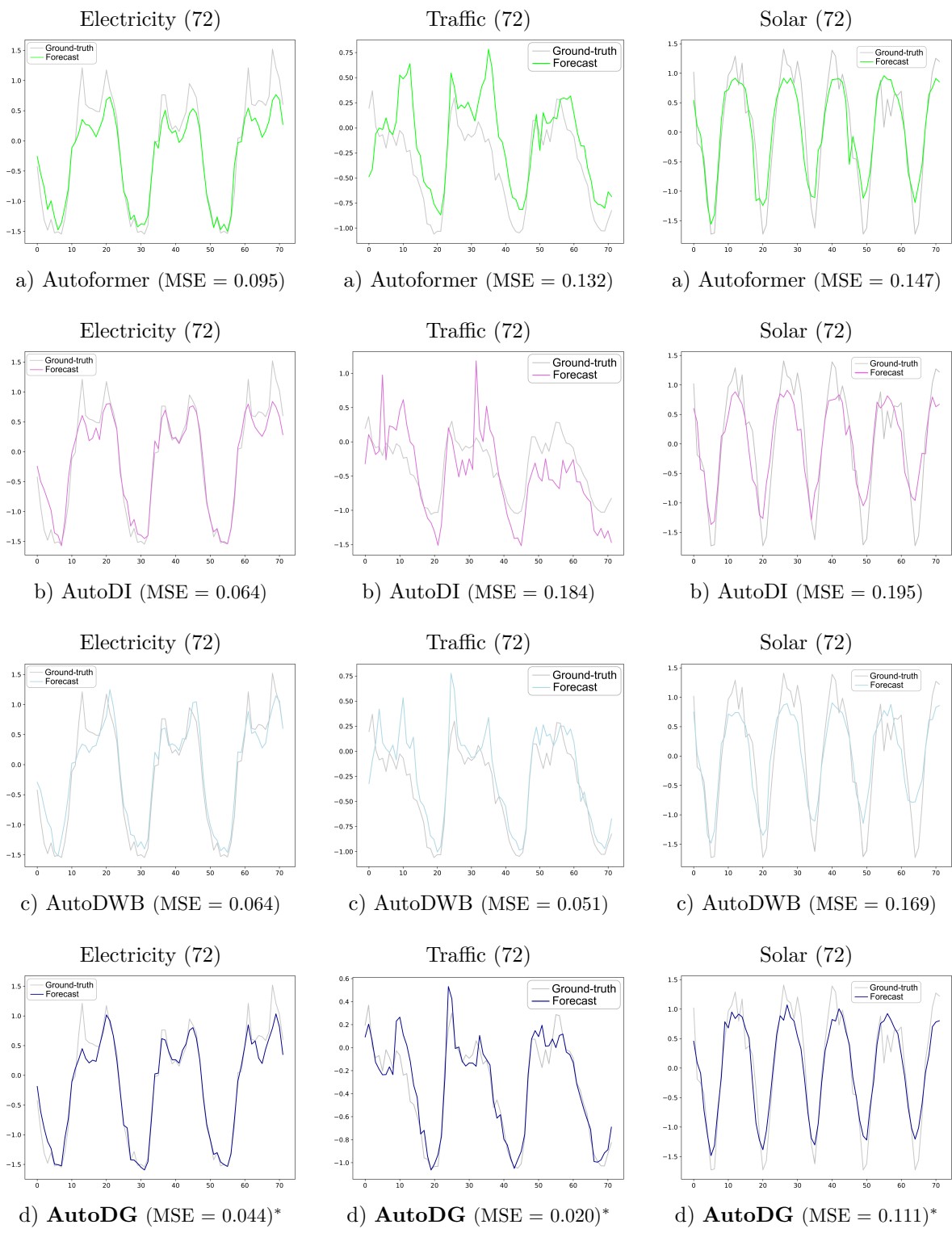

Figure 3: Example forecasts of four treatments of autoformer model on Traffic, Electricity, and Solar datastes for 72 future time steps. The values are plotted in Z-score normalized space. Best MSE results for each column (dataset) are denoted with the ∗ symbol.

in terms of MSE. All forecasting models are evaluated on their ability to predict for the next 24, 48, 72 and 96 future time steps. Note that the results of baseline models are included in both tables for the readability purposes. When treating the Autoformer and Informer models, our proposed GP-based forecast-blur-denoise forecasting model predominately outperforms the AutoDI/InfoDI, and the initial forecasting models. This finding is consistent across all datasets and depicted in Figure 3.

Next we conduct an ablation study by comparing our approach with several canonical denoising and boosting approaches. Table 3 and 4 presents an overview of the additional treatments applied to the Autoformer and Informer model. Compared to using a denoising approach during training (AutoDT/InfoDT), our model consistently performs better, further emphasizing the effectiveness of our end-to-end forecast-blur-denosie model. When attempting to denoise predictions directly without applying corruption, AutoDWC does not consistently yield improvements over the untreated forecasting model Autoformer. This highlights the significance of our GP blur model in enhancing the accuracy of forecasts. This inconsistency in improving the untreated forecasting model is apparent in the AutoDI model as well. This reinforces our hypothesis that denoising isotropic noise does not provide benefits for time series data. This applies when dealing with time series forecasting model that do not exhibit jitters in their predictions.

Figure 3 illustrates forecasts for 72 future time steps of one sample of the hold-out test of all datasets obtained by four treatments of the Autoformer forecasting model. These treatments include a) the initial untreated forecasting model, Autoformer a), b) AutoDI c) AutoAWB, and d) our AutoDG. The overall trend in the forecast patterns indicates that the standalone Autoformer model a) generally tracks the ground-truth, albeit fine-grained features are only roughly reproduced. AutoDI b) introduces jitters leading to inaccurate local behavior. AutoDWB c) fails to accurately reproduce fine-grained features including details and extreme values, even though it yields lower MSE on the Traffic dataset, compared to a) and b). Our proposed model with GP blurring and denoising (AutoDG) d) produces the most accurate forecasts by accurately predicting coarse-grained behavior of peaks and valleys, as well as fine-grained behavior such as smooth slopes, details and improved extreme values prediction.

The success of our forecast-blur-denoise model in comparison to traditional denoising models lies in the division of responsibilities arising from the GP blur model. As a result the initial forecasting model focuses on predicting the broader patterns and trends, while a dedicated denoising forecaster addresses the finer details. The outcome is an overall refined forecasting model that excels in both coarse and fine-grained forecasting tasks.

Please refer to Appendix A for the complementary results of the MAE metric and additional ablation studies.

## 4 Conclusion

In this paper, we study the multi-horizon time series forecasting problem and propose an end-to-end forecast-blur-denoise framework. By end-to-end training the parameters of a GP blur model, we encourage a separation of concerns, leading the initial forecasting model to focus on accurately predicting the coarse-grained behavior, while the denoising model is filling in the fine-grained details. This ultimately leads to an enhanced coarse and fine-grained forecasting methods. In addition to end-to-end training of the multi-component model, we utilize methods from variational techniques to guide the training of the GP blur model for achieving the ideal noise model.

Where most denoising methods leverage isotropic Gaussians, we hypothesize that a blur/noise model that generates smooth and correlated functions offers the most advantages in the time series domain. In line with Robinson et al. (2018), we find that using a noise model with temporal correlation, such as the GPs, is advantageous over uncorrelated noise models. Our experiments show that our proposed framework with GP blur model is significantly outperforming the forecasting model without any denoising as well as denoising isotropic Gaussian noise. Additionally, we show that our forecast-blur-denoise framework predominately outperforms an approach that uses blurring and denoising only during training (AutoDT).

A strength of our approach is that it can be applied to any neural forecasting model. We demonstrate the effectiveness of our approach across three real-world datasets, different horizons, and several the-state-of-the-art

forecasting models, including Autoformer and Informer. In all experiments, our proposed forecast-blur-denoise forecasting approach with GPs leads to significant improvements.

## Broader Impact Statement

Time series forecasting offers societal benefits such as optimizing traffic lights, aligning energy grid capacities with solar power, and providing quantitative models for scientific understanding. However, like any foundational technology, it can also be misused for negative impacts, like aiding criminals or misinformation campaigns. By enhancing machine learning methods for more accurate forecasting, we aim to improve current practices, with the hope that the positive aspects will prevail.

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

# A  Appendix

Here, we provide the quantitative results in terms of MAE evaluation metric. Table 5 and 7 include the MAE results of the initial three treatments and other baseline models. Note that results of the baseline models are repeated for the purpose of readability. Additionally, the MAE results of the ablation studies of other denoising/boosting models are reported in Table 6 and 8.

To show that the improvement of our proposed model indeed stems from its ability and not the higher number of parameters regarding the GP model, we conducted an ablation study by increasing the number of layers (parameters) of the standalone forecasting models and included the results in terms of both MSE and MAE in Table 9 and 10.

Table 5: Overall results of the quantitative evaluation of our forecast-blur-denoise and other baseline models in terms of average and standard error of **MAE**. We compare the forecasting models on all three datasets with different number of forecasting steps. A lower **MAE** indicates a better model. Our forecast-blur-denoise with GPs enhances the performance of the original Autoformer and isotropic Gaussian noise model (AutoDI). Note that for a fair comparison, all baseline models share the same experimental setup as our proposed model. Reported results may differ from the original baseline papers, and the baseline models are available in our online repository.

| Dataset | Horizon | AutoDG(Ours) | Autoformer | AutoDI | NBeats | DLinear | DeepAR | CMGP | ARIMA |
|---------|---------|--------------|------------|--------|--------|---------|--------|------|-------|
| Traffic | 24 | **0.333** ±0.010 | **0.334** ±0.007 | **0.340** ±0.007 | 0.384 ±0.001 | 0.447 ±0.000 | 0.652 ±0.000 | 0.645 ±0.000 | 0.770 ±0.000 |
| | 48 | **0.328** ±0.001 | 0.368 ±0.006 | 0.343 ±0.006 | 0.408 ±0.003 | 0.462 ±0.000 | 0.650 ±0.000 | 0.642 ±0.000 | 0.776 ±0.000 |
| | 72 | **0.358** ±0.013 | **0.356** ±0.003 | **0.356** ±0.005 | 0.413 ±0.004 | 0.466 ±0.000 | 0.636 ±0.000 | 0.648 ±0.000 | 0.782 ±0.000 |
| | 96 | **0.333** ±0.000 | 0.359 ±0.004 | 0.366 ±0.004 | 0.414 ±0.002 | 0.471 ±0.000 | 0.632 ±0.000 | 0.647 ±0.000 | 0.773 ±0.000 |
| Electricity | 24 | **0.249** ±0.001 | 0.265 ±0.003 | 0.258 ±0.001 | 0.294 ±0.004 | 0.299 ±0.000 | 0.862 ±0.000 | 0.840 ±0.000 | 0.959 ±0.000 |
| | 48 | **0.275** ±0.003 | 0.292 ±0.007 | 0.301 ±0.003 | 0.310 ±0.007 | 0.308 ±0.000 | 0.853 ±0.000 | 0.839 ±0.000 | 0.971 ±0.000 |
| | 72 | **0.303** ±0.004 | **0.297** ±0.006 | **0.303** ±0.004 | 0.322 ±0.007 | 0.323 ±0.000 | 0.856 ±0.000 | 0.836 ±0.000 | 0.987 ±0.000 |
| | 96 | **0.304** ±0.001 | 0.372 ±0.010 | 0.325 ±0.002 | 0.324 ±0.002 | 0.329 ±0.000 | 0.850 ±0.000 | 0.832 ±0.000 | 0.983 ±0.000 |
| Solar | 24 | **0.548** ±0.009 | 0.603 ±0.002 | 0.574 ±0.008 | 0.632 ±0.001 | 0.801 ±0.000 | 0.885 ±0.000 | 0.885 ±0.000 | 1.100 ±0.000 |
| | 48 | **0.612** ±0.003 | 0.656 ±0.003 | 0.638 ±0.003 | 0.710 ±0.001 | 0.864 ±0.000 | 0.865 ±0.000 | 0.888 ±0.000 | 1.102 ±0.000 |
| | 72 | **0.702** ±0.0001 | 0.729 ±0.017 | **0.702** ±0.001 | 0.744 ±0.001 | 0.894 ±0.000 | 0.873 ±0.000 | 0.885 ±0.000 | 1.106 ±0.000 |
| | 96 | **0.725** ±0.000 | 0.754 ±0.009 | 0.747 ±0.005 | 0.781 ±0.002 | 0.911 ±0.000 | 0.866 ±0.000 | 0.882 ±0.000 | 1.097 ±0.000 |

Table 6: Comparison of different denoising baselines to our proposed model when treating the Autoformer model. We find that our approach predominantly outperforms the other denoising approaches. Results are reported as average and standard error of **MAE**. A lower **MAE** indicates a better forecasting model.

| Dataset | Horizon | AutoDG(Ours) | Autoformer | AutoDI | AutoDWC | AutoRB | AutoDT |
|---------|---------|--------------|------------|--------|---------|--------|--------|
| Traffic | 24 | **0.333**±0.010 | **0.334**±0.007 | **0.340**±0.007 | 0.345±0.009 | 0.391±0.005 | 0.349±0.005 |
| | 48 | **0.328**±0.001 | 0.368±0.006 | 0.343±0.006 | 0.351±0.005 | 0.359±0.002 | 0.361±0.016 |
| | 72 | **0.358**±0.013 | **0.356**±0.003 | **0.356**±0.005 | 0.361±0.002 | 0.383±0.005 | 0.379±0.001 |
| | 96 | **0.304**±0.000 | 0.325±0.004 | 0.372±0.004 | 0.362±0.004 | 0.368±0.005 | 0.379±0.005 |
| Electricity | 24 | **0.249**±0.001 | 0.265±0.003 | 0.258±0.001 | 0.272±0.001 | 0.380±0.001 | 0.263±0.007 |
| | 48 | **0.275**±0.003 | 0.292±0.007 | 0.301±0.003 | 0.306±0.002 | 0.311±0.002 | 0.288±0.003 |
| | 72 | **0.303**±0.004 | **0.297**±0.006 | **0.303**±0.004 | **0.295**±0.009 | 0.330±0.021 | **0.305**±0.003 |
| | 96 | **0.304**±0.001 | 0.372±0.010 | 0.325±0.002 | 0.324±0.005 | 0.386±0.005 | 0.318±0.005 |
| Solar | 24 | **0.548**±0.009 | 0.603±0.002 | 0.574±0.008 | 0.549±0.008 | 0.601±0.005 | 0.598±0.006 |
| | 48 | **0.612**±0.003 | 0.656±0.003 | 0.638±0.003 | 0.656±0.002 | 0.645±0.003 | 0.655±0.003 |
| | 72 | **0.702**±0.001 | 0.729±0.017 | **0.702**±0.001 | 0.724±0.008 | 0.720±0.010 | 0.709±0.004 |
| | 96 | **0.725**±0.000 | 0.754±0.009 | 0.747±0.005 | 0.746±0.006 | 0.738±0.007 | 0.745±0.006 |

Table 7: Overall results of the quantitative evaluation of our forecast-blur-denoise and other baseline models in terms of average and standard error of **MAE**. We compare the forecasting models on all three datasets with different number of forecasting steps. A lower **MAE** indicates a better model. Our forecast-blur-denoise with GPs enhances the performance of the original Informer and isotropic Gaussian noise model (InfoDI). Note that for a fair comparison, all baseline models share the same experimental setup as our proposed model. Reported results may differ from the original baseline papers, and the baseline models are available in our online repository.

| Dataset | Horizon | InfoDG(Ours) | Informer | InfoDI | NBeats | DLinear | DeepAR | CMGP | ARIMA |
|---|---|---|---|---|---|---|---|---|---|
| Traffic | 24 | 0.355 ±0.007 | **0.329** ±0.006 | 0.342 ±0.004 | 0.384 ±0.001 | 0.447 ±0.000 | 0.652 ±0.000 | 0.645 ±0.000 | 0.770 ±0.000 |
| | 48 | **0.350** ±0.001 | **0.354** ±0.012 | **0.362** ±0.005 | 0.408 ±0.003 | 0.462 ±0.000 | 0.650 ±0.000 | 0.642 ±0.000 | 0.776 ±0.000 |
| | 72 | **0.345** ±0.003 | 0.377 ±0.002 | 0.353 ±0.008 | 0.413 ±0.004 | 0.466 ±0.000 | 0.636 ±0.000 | 0.648 ±0.000 | 0.782 ±0.000 |
| | 96 | **0.397** ±0.004 | **0.402** ±0.011 | **0.402** ±0.005 | **0.414** ±0.002 | 0.471 ±0.000 | 0.632 ±0.000 | 0.647 ±0.000 | 0.773 ±0.000 |
| Electricity | 24 | **0.290** ±0.008 | **0.300** ±0.005 | **0.298** ±0.003 | **0.294** ±0.004 | **0.299** ±0.000 | 0.862 ±0.000 | 0.840 ±0.000 | 0.959 ±0.000 |
| | 48 | **0.311** ±0.002 | 0.349 ±0.009 | 0.325 ±0.002 | 0.310 ±0.007 | 0.308 ±0.000 | 0.853 ±0.000 | 0.839 ±0.000 | 0.971 ±0.000 |
| | 72 | **0.345** ±0.003 | 0.377 ±0.002 | 0.353 ±0.008 | 0.322 ±0.002 | 0.323 ±0.000 | 0.856 ±0.000 | 0.836 ±0.000 | 0.987 ±0.000 |
| | 96 | **0.342** ±0.004 | 0.378 ±0.011 | 0.379 ±0.005 | 0.324 ±0.002 | 0.329 ±0.000 | 0.850 ±0.000 | 0.832 ±0.000 | 0.983 ±0.000 |
| Solar | 24 | **0.533** ±0.005 | 0.597 ±0.002 | 0.563 ±0.003 | 0.632 ±0.001 | 0.801 ±0.000 | 0.885 ±0.000 | 0.885 ±0.000 | 1.100 ±0.000 |
| | 48 | **0.624** ±0.007 | 0.681 ±0.013 | 0.635 ±0.005 | 0.710 ±0.001 | 0.864 ±0.000 | 0.865 ±0.000 | 0.888 ±0.000 | 1.102 ±0.000 |
| | 72 | **0.690** ±0.013 | 0.752 ±0.017 | 0.735 ±0.019 | 0.744 ±0.001 | 0.894 ±0.000 | 0.873 ±0.000 | 0.885 ±0.000 | 1.106 ±0.000 |
| | 96 | **0.727** ±0.006 | 0.772 ±0.012 | 0.764 ±0.004 | 0.781 ±0.002 | 0.911 ±0.000 | 0.866 ±0.000 | 0.882 ±0.000 | 1.097 ±0.000 |

Table 8: Comparison of different denoising baselines to our forecast-blur-denoise approach with GPs when treating Informer forecasting model. We find that our approach predominantly outperforms the other denoising approaches. Results are reported as average and standard error of **MAE**. A lower **MAE** indicates a better forecasting model.

| Dataset | Horizon | InfoDG(Ours) | Informer | InfoDI | InfoDWC | InfoRB | InfoDT |
|---|---|---|---|---|---|---|---|
| Traffic | 24 | 0.355±0.007 | **0.329**±0.006 | 0.342±0.004 | 0.337±0.003 | 0.331±0.003 | 0.379±0.003 |
| | 48 | **0.345**±0.001 | 0.377±0.013 | 0.353±0.005 | 0.348±0.003 | 0.353±0.009 | 0.375±0.006 |
| | 72 | **0.345**±0.003 | 0.377±0.010 | 0.353±0.004 | 0.348±0.001 | 0.379±0.005 | 0.361±0.006 |
| | 96 | **0.350**±0.004 | **0.354**±0.006 | **0.362**±0.007 | **0.348**±0.008 | 0.379±0.005 | **0.361**±0.011 |
| Electricity | 24 | **0.290**±0.008 | **0.300**±0.005 | **0.298**±0.003 | **0.295**±0.004 | **0.302**±0.009 | 0.318±0.003 |
| | 48 | **0.311**±0.002 | 0.349±0.009 | 0.325±0.002 | 0.333±0.004 | 0.343±0.009 | 0.343±0.005 |
| | 72 | **0.336**±0.003 | 0.371±0.002 | 0.359±0.008 | 0.362±0.008 | 0.359±0.006 | 0.367±0.008 |
| | 96 | **0.342**±0.004 | 0.378±0.0011 | 0.379±0.005 | 0.384±0.006 | 0.375±0.001 | 0.370±0.007 |
| Solar | 24 | **0.533**±0.005 | 0.597±0.002 | 0.563±0.003 | 0.551±0.001 | 0.573±0.008 | 0.596±0.007 |
| | 48 | **0.624**±0.007 | 0.681±0.013 | 0.635±0.005 | 0.649±0.011 | 0.675±0.012 | 0.681±0.012 |
| | 72 | **0.690**±0.013 | 0.752±0.017 | 0.735±0.019 | 0.736±0.011 | 0.763±0.020 | 0.735±0.004 |
| | 96 | **0.727**±0.006 | 0.772±0.012 | 0.764±0.004 | 0.753±0.005 | 0.777±0.014 | 0.766±0.002 |

Table 9: Comparison of our forecast-blur-denoise approach with standalone forecasting models with higher number of layers (parameters) denoted by † sign. Initially, the number of layers for our proposed model and other baselines are chosen from $\{1, 2\}$, however to show that the performance of our model indeed stems from its mechanism, we included the results of standalone forecasting models with number of layers chosen from $\{3, 4\}$. Results are reported as average and standard error of **MSE**. A lower **MSE** indicates a better forecasting model.

| Dataset | Horizon | **AutoDG(Ours)** | Autoformer | Autoformer† | **InfoDG(Ours)** | Informer | Informer† |
|---|---|---|---|---|---|---|---|
| Traffic | 24 | 0.392 ±0.006 | 0.412 ±0.006 | **0.359** ±0.007 | **0.398** ±0.006 | 0.421 ±0.006 | 0.422 ±0.009 |
| | 48 | 0.387 ±0.001 | 0.422 ±0.007 | **0.383** ±0.001 | **0.399** ±0.001 | 0.434 ±0.001 | 0.486 ±0.010 |
| | 72 | **0.380** ±0.001 | **0.383** ±0.002 | 0.442 ±0.006 | **0.380** ±0.001 | 0.436 ±0.001 | 0.412 ±0.003 |
| | 96 | **0.385** ±0.003 | 0.400 ±0.004 | 0.416 ±0.001 | **0.397** ±0.003 | **0.402** ±0.003 | 0.408 ±0.005 |
| Electricity | 24 | **0.165** ±0.001 | 0.187 ±0.003 | 0.242 ±0.007 | **0.193** ±0.001 | 0.222 ±0.001 | 0.266 ±0.001 |
| | 48 | **0.188** ±0.003 | 0.203 ±0.008 | 0.232 ±0.005 | **0.222** ±0.002 | 0.262 ±0.002 | 0.293 ±0.002 |
| | 72 | **0.209** ±0.004 | 0.230 ±0.001 | 0.263 ±0.004 | **0.238** ±0.001 | 0.280 ±0.003 | 0.310 ±0.002 |
| | 96 | **0.211** ±0.001 | 0.230 ±0.014 | 0.224 ±0.004 | **0.242** ±0.001 | 0.289 ±0.002 | 0.327 ±0.003 |
| Solar | 24 | **0.446** ±0.002 | 0.472 ±0.003 | 0.524 ±0.001 | **0.455** ±0.009 | 0.524 ±0.003 | 0.498 ±0.001 |
| | 48 | **0.546** ±0.005 | 0.603 ±0.003 | 0.622 ±0.001 | **0.556** ±0.005 | 0.629 ±0.003 | 0.690 ±0.031 |
| | 72 | **0.666** ±0.003 | **0.667** ±0.004 | 0.701 ±0.004 | **0.643** ±0.003 | 0.729 ±0.023 | 0.716 ±0.024 |
| | 96 | **0.713** ±0.004 | 0.739 ±0.009 | 0.744 ±0.002 | **0.708** ±0.004 | 0.770 ±0.004 | 0.738 ±0.002 |

Table 10: Comparison of our forecast-blur-denoise with standalone forecasting models with higher number of layers (parameters) denoted by † sign. Initially, the number of layers for our proposed model and other baselines are chosen from $\{1, 2\}$, however to show that the performance of our model indeed stems from its mechanism, we included the results of standalone forecasting models with number of layers chosen from $\{3, 4\}$. Results are reported as average and standard error of **MAE**. A lower **MAE** indicates a better forecasting model.

| Dataset | Horizon | **AutoDG(Ours)** | Autoformer | Autoformer† | **InfoDG(Ours)** | Informer | Informer† |
|---|---|---|---|---|---|---|---|
| Traffic | 24 | **0.333** ±0.010 | **0.334** ±0.007 | **0.332** ±0.002 | 0.355 ±0.007 | **0.329** ±0.006 | 0.382 ±0.003 |
| | 48 | **0.328** ±0.001 | 0.368 ±0.002 | 0.336 ±0.001 | **0.345** ±0.001 | 0.377 ±0.013 | 0.402 ±0.005 |
| | 72 | **0.358** ±0.013 | **0.356** ±0.003 | 0.357 ±0.001 | **0.345** ±0.003 | 0.377 ±0.010 | 0.382 ±0.011 |
| | 96 | **0.385** ±0.003 | 0.400 ±0.004 | 0.370 ±0.001 | **0.397** ±0.003 | **0.402** ±0.002 | 0.413 ±0.003 |
| Electricity | 24 | **0.249** ±0.001 | 0.265 ±0.003 | 0.303 ±0.003 | **0.290** ±0.003 | **0.300** ±0.006 | 0.332 ±0.002 |
| | 48 | **0.275** ±0.001 | 0.292 ±0.007 | 0.317 ±0.002 | **0.311** ±0.002 | 0.349 ±0.009 | 0.377 ±0.002 |
| | 72 | **0.303** ±0.004 | **0.297** ±0.007 | 0.351 ±0.002 | **0.336** ±0.003 | 0.371 ±0.002 | 0.384 ±0.002 |
| | 96 | **0.304** ±0.001 | 0.372 ±0.010 | 0.317 ±0.001 | **0.342** ±0.004 | 0.378±0.001 | 0.411 ±0.003 |
| Solar | 24 | **0.548** ±0.009 | 0.603 ±0.002 | 0.608 ±0.001 | **0.533** ±0.005 | 0.597 ±0.002 | 0.575 ±0.000 |
| | 48 | **0.612** ±0.003 | 0.656 ±0.003 | 0.672 ±0.004 | **0.624** ±0.007 | 0.681 ±0.013 | 0.745 ±0.020 |
| | 72 | **0.702** ±0.001 | 0.729 ±0.017 | 0.707 ±0.001 | **0.690** ±0.013 | 0.752 ±0.017 | 0.762 ±0.016 |
| | 96 | **0.725** ±0.000 | 0.754 ±0.009 | 0.737 ±0.002 | **0.727** ±0.006 | 0.772 ±0.012 | 0.785 ±0.010 |

