# OpenReview forum: "Coarse and Fine-grained Forecasting Via Gaussian Process Blurring Effect"
_TMLR — Rejected by TMLR_

### Review · Reviewer_8tKK · 2024-03-12

**Summary Of Contributions:**

The paper proposes a method for time series forecasting where the goal is, given some temporal data, predicting the continuation of the sequence. The main contribution of the method is that an initial coarse forecasting gets refined with a blurring and denoising step. As a result, the method can be used on top of any time series forecasting method. The method has been evaluated on top of two methods, as well as compared to other forecasting methods for 3 univariate datasets.

**Audience:**

Yes

**Broader Impact Concerns:**

Not needed mroe than beyond what is already provided in the paper

**Claims And Evidence:**

No

**Requested Changes:**

- Clarifying the relation between motivation, used terminology, and proposed method.
- Ablating the role of the blur step.
- Given that the method is claimed that it can be used with any time series forecasting model, it would be interesting to see the effect on more than just two baselines.

**Strengths And Weaknesses:**

**Strengths**
- The proposed method can be used on top of any existing model leading to improved results. This has been shown for two baseline methods.
- Tested two noise models.


**Weaknesses**

How the method is actually implemented and how it relates to the motivation was confusing to the reviewer:
- Blurring the prediction leads to a more jittery curve in the motivational figure 1 which seems confusing.
- The motivation of denoising is to obtain fine-grained details, but Fig.1 shows that the denoising smooths the blurred prediction. What is meant with fine-grained details?
- In the intro, timestep t_0 is part of the historical time series X and of the prediction Y. Notation should be clarified.

Experiments section can benefit from more details and additional ablations:
- According to the methodology section, the method can be applied to any time series forecasting method. However, it is only shown on top of transformer methods. Is it specific that it only works with transformers or would it also work with other baselines?
- As denoising model the same architecture as of the initial forecasting model is used. In between the blur model is used. How important is that step of deblurring, i.e., how is the performance if only step 1 and 3 are performed (without the blurring step)?
- For comparisons the baseline methods are evaluated in the same experimental setup as the proposed method but that might differ from the original paper, which makes sense for the comparisons. But for a fair reporting of the previous work, also the original performance of the baselines should be reported. And adding how the setup has been changed.
- Over how many seeds are the numbers in table 2 computed?
- Some hyper-parameter optimization has been performed. However, the reporting of the final used setting is missing.

---

### Review · Reviewer_A31x · 2024-03-21

**Summary Of Contributions:**

This paper introduces a new method for time series forecasting based on combining a forecasting model with a diffusion model. The essential idea is that diffusion can help to recover fine details of output signals. The approach is evaluated on several datasets and compared to several baselines.

**Audience:**

Yes

**Broader Impact Concerns:**

Time series forecasting is an important problem. It is related to generative modeling. Thus, combining TSF with diffusion models may lead to interesting findings.

**Claims And Evidence:**

No

**Requested Changes:**

See above

**Strengths And Weaknesses:**

One strength of this approach is its simplicity. Additionally, the paper is easy to follow and understand.

Unfortunately, there are many shortcomings in my opinion. First, the paper does not motivate why combining forecasting and diffusion "makes sense". The repeated intuition about coarse vs. fine details is not convincing. Second, the related work section is insufficient. The authors should cover relevant works on forecasting, generative modeling, and other related literature. Third, the evaluation section considers Autoformer and Informer as baselines, however, these approaches are far from being SOTA. Fourth, the authors change the standard long-term forecasting task where others consider a lookback of 96 (or higher) and prediction horizons of 96,192,336 and 720. Fifth, the authors propose to change the noise distribution of diffusion models (which is an interesting idea), however, they do not re-derive diffusion equations. However, new prior distributions require a re-derivation. Sixth, the losses in Eq. 2 are not described well.

---

### Review · Reviewer_8tsp · 2024-03-24

**Summary Of Contributions:**

The paper presents a novel multistage DNN framework for time series forecasting which takes a backbone forecasting model to produce an initial forecast which is then blurred using a Gaussian Process blurring model. The finer grained details are then restored using a deblurring model. The entire multistage system is trained end-to-end. Experiments are performed on 3 datasets and compared with several models from the art to demonstrate that the performance of the backbone model can be improved to get results surpassing the state of the art. It is argued in the paper that such a multistage setup – forecasting, blurring, and then deblurring (a) allows for a principled ‘division-of-labor’ responsible for improved performance, and (b) can be used with any backbone forecasting model, improving its performance.

**Audience:**

Yes

**Broader Impact Concerns:**

A ‘broader impact’ statement is included in the paper. I don’t see any concerns that need to be addressed beyond the present statement.

**Claims And Evidence:**

No

**Requested Changes:**

In my view, the research underlying the paper is not mature and close to publication. The technical understanding of the problem, the prior art, the unmet need and addressing that via a novel approach and its experimental validation – everything needs to go back to the drawing board.

The critique above can be used to make appropriate changes. However, since they would need to be comprehensive, I don't see the need to individually list them down.

(I may appear to be overly harsh, but the criticism is well-meaning. I hope it is taken constructively and will lead to good research from the authors in the future).

**Strengths And Weaknesses:**

**Strengths**

The paper takes on an important problem – time series forecasting – which is relevant across a large variety of application domains.

A multistage architecture inspired by different granularity (scale) of information and its import on forecasting the future, is a good direction for research, which this paper undertakes. (Although, I’m not convinced of the technical reasoning behind the architecture design - more below).

**Weaknesses**

There are multiple, critical, weaknesses in all aspects of the presented work. I list the major ones below instead of comprehensively listing all:

**Solution Design**: The reasoning employed to design the solution is rather weak.

(a) While one may argue that the ‘Markovian window’ varies for information in the signal at different granularities, the proposed solution in which two different scales of information are suggested where the coarser information is dependent on the past while the finer is not, seems arbitrary and ad hoc.

(b) Secondly, the blurring operation is lossy and loses any important high-resolution information in the forecast that the forecasting model may have predicted. The blurring itself seems arbitrary and ad hoc.

**Weak Reasoning**: The whole discussion on isotropic noise vs Gaussian processes misses the role such signal corrupting approaches play in the training of a generative model and the signal properties it may learn.

**Technical terminology**: Using denoising for deblurring is jarring.

**Experimental validation** is inadequate.

Experimental Settings: The baselines used – DLinear, Informer, and Autoformer evaluate performance (a) in both univariate and multivariate settings while the paper only evaluates in the simpler univariate setting, (b) over longer forecasting horizons (going as much as 7.5x longer than those considered in this paper), (c) on more datasets and benchmarks than the ones used in the paper (3). The important benchmark ETT* is not considered at all.

Reproduction of prior art: I checked the quantitative performance in the tables in the DLinear, Informer, and Autoformer papers. I see a large difference in the results of the prior art as reported in those papers in comparison to the numbers presented by the authors, often to the detriment of the prior art. I don’t understand the reasons for the large difference which is also not explained in the paper. Note that a lot of prior art has source code available.

Validation of the proposed approach: Consequently, it is hard to draw any conclusions about the claimed strengths of the proposed approach.

---

### Comment · Action_Editor_5fs8 · 2024-04-06
**Discussion period under way**

Dear authors,

All three reviewers have provided reviews for your paper. Accordingly, please respond to their comments and questions.

A.E.

---

### Decision · Action_Editor_5fs8 · 2024-04-26

**Recommendation:** Reject

**Comment:**

All reviewers agree that there are major drawbacks in the manuscript that have not been addressed. These limitations include:

1) Methodological: Parts of the proposed approach is not clear to the reviewers. E.g., reviewers were confused about the integration and motivation for the blurring step. For example, why is this lossy process needed, and why isn’t it detrimental to lose the high frequency information that might be useful for prediction? The implications of this blurring effect in Fig. 1 are also unclear. Furthermore, the analytical implications of employing a different sampling distribution for the diffusion models are not addressed.

2) Lack of background on necessary related work: Reviewers noted that references to modern forecasting approach are missing, and many of the used methods are not state-of-the-art.

3) Numerical results and insufficiency of evidence: It is unclear what the empirical benefits of the blurring steps are; there are important relevant benchmarks considered in related works that are not explored in this manuscript. Furthermore, a reviewer pointed to discrepancies between performance metrics reported in this paper by prior art vs those reported in the respective original publications.

Lastly, the authors did not respond to any of the comments and questions by the reviewers.

**Audience:**

Some audience in TMLR would be interested in this work, if it were ready for publication.

**Claims And Evidence:**

All reviewers agree that the claims made in their manuscript is not well supported by numerical evidence nor justified.